# Magnitude and predictors of HIV-Drug resistance in Africa: A protocol for systematic review and meta-analysis

**Mulugeta Melku**[1,2]*, **Hailay Abrha Gesesew**[3,4], **Paul R. Ward**[4]

**1** School of Biomedical and Laboratory Sciences, College of Medicine and Health Sciences, University of Gondar, Gondar, Ethiopia, **2** College of Medicine and Public Health, Flinders University, Adelaide, South Australia, Australia, **3** School of Public Health, Mekelle University, Mekelle, Ethiopia, **4** Research Centre for Public Health Policy, Torrens University, Adelaide, South Australia, Australia

* mulugeta.melku@gmail.com, gobe0011@flinders.edu.au

**Funding:** The author(s) received no specific funding for this work.

**Competing interests:** The authors declared that there is no competing interests exist.

## Abstract

### Introduction

Human Immunodeficiency Virus (HIV) is continued to be a major public health problem in low-income countries and more importantly in Africa. For the last decade, access to Antiretroviral Therapy (ART) and its impact in improving quality of life and reducing HIV-related morbidity and mortality has significantly been improved in Africa. Nevertheless, the emergency of HIV drug resistance (HIVDR) has posed challenges in achieving optimal ART treatment outcomes and is alarmingly increasing globally in general and in Africa in particular. Comprehensive epidemiological data on the magnitude of HIVDR and HIVDR mutations, and predictors of HIVDR are, however, limited in Africa.

### Objective

The main objective of this systematic review will be to estimate the pooled proportion of HIVDR and HIVDR mutations, and identify factors associated with HIVDR among people living with HIV/AIDS (PLWH) in Africa.

### Method

Published Literature from 2000 until 30 October 2021 will be searched in PubMed/Medline Ovid, HINARI, SCOPUS, EMBASE, CINAHL, Web of Sciences, and Cochrane electronic databases. Initially, the literature will be screened based on title/abstract and followed by full-text appraisal for methodological quality using JBI critical appraisal tools. Data will be extracted from eligible articles after the full-text appraisal. Heterogeneity will be qualitatively assessed by a visual Funnel plot and quantitatively measured by an index of heterogeneity ($I^2$ statistics). Random-effects model will be fitted to estimate the proportion of HIVDR and each HIVDR mutations. Sub-group and sensitivity analyses will be conducted to reduce heterogeneity. Meta-regression will be done by median year of sampling per study to observe the pattern of changes over time. Publication bias will be assessed by egger's statistics. In

case of publication bias, Trim and Fill analysis will be conducted to overcome small-study effect. Data analysis will be performed using Stata version 14.

## Ethics and dissemination

As the data sources are published papers, the protocol will not require an ethical approval letter. The final report of the review will be published in a peer-reviewed journal.

## Introduction

The human immunodeficiency virus (HIV) continues to be a major public health problem globally. At the end of 2020, 37.7 million (95% uncertainty level: 30.2–45.1 million) people were living with HIV, more than two-thirds of whom (25.4 million) were in the African World health organization (WHO) region [1]. Moreover, one and half million people were newly infected, and more than half a million people died from HIV-related causes in 2020 [1]. Nearly three-fourth of PLWH had access to ART [2,3]. The Global Burden of Disease estimated that HIV/AIDS was the eleventh common cause of disability-adjusted Life-years globally in 2019 – the second top cause for 25–49 years old, and the ninth for 10–24 years old [4].

For the last decade, the introduction and scale-up of ART have been significantly and positively improved the quality of life, life expectancy, and reduced mortality in resource-limited countries including Africa [5–8]. However, the emergence of HIVDR becomes the growing concern and challenge of HIV care, as it reduces the effectiveness of ART success, and leads to morbidity and mortality despite expanding the HIV-care services [9].

HIV drug resistance has the potential to be a major public health problem. The WHO reported that up to 26% of people initiating treatment are infected with a virus carrying resistance to first-line antiretroviral drugs [10]. The level of HIVDR is higher in infants born to mothers infected with HIV, which reaches up to 69% [10,11]. In Africa, its prevalence has been raising [12,13]. According to the 2019 WHO report, in six African countries that participated in national HIVDR surveillance, the prevalence is high in children and adults. Particularly, the magnitude of resistance to non-nucleoside reverse transcriptase inhibitors (NNRTI) is high [14], partly because 1) NNRTI-based regimens are frequently prescribed in low-income countries than in high-income countries as these regiment have longer half-life and durability [15], and 2) NNRTI regiments have a low genetic barrier to resistance [16]. Studies also reported that the HIVDR is found to be a problem among people who are ART-naïve starting first-line ART [17,18]. This situation limits the treatment outcome in individuals who are on the first-line ART regimen [19]. The resistance to first-line ART drugs would also affect effectiveness of second-line ART drugs [19,20]. And more recently, there are evidence revealing a high rate of drug resistance compromised the efficacy of second-line ART regimens. Alarmingly, the burden of HIVDR is more prevalent in those individuals who are on second-line ART than those who are on the first-line ART regimen [19]. A retrospective study conducted in Zimbabwe on adolescents and adults who failed the first-line ART regimens demonstrated a high level of drug resistance mutations despite enhanced adherence counselling and switching to the second-line regiment [20]. Moreover, a study done in Namibia demonstrated that there was a high level of HIVDR among adults failing the second-line ART regimen [21]. Notably, for those PLWH who are on the second-line ART regimen and failings either the first-line or second-ART regimens, high level of viral load constitutes viral strains harboring non-drug associated genetic changes that could predispose to high rate of HIVDR development [22]. In

addition, a delayed in switching to and unnecessary switching to the second-line regimens [23,24] could lead to the accumulation drug resistance mutations.

A change in the genetic structure of HIV because of mutations is the main cause of HIVDR. The genetic structural changes of the virus affect the ability of antiretroviral drugs to block the replication of HIV. Several types of genetic mutations associated with HIVDR has been documented in Africa [21,25–30]. M184V, D67N, K65R and K70R are the commonest nucleoside reverse transcriptase inhibitor (NRTI) resistance mutations that affect the effectiveness of antiretroviral drugs acting on reverse transcriptase machinery of the HIV [26–33]. Similarly, K103N, Y181C, V106M, and G190A mutations are frequently reported to have associated with NNRTIs drug resistance [25–29,31–33]. Some mutations like M184V, M184IV, and D67NS are found in NRTIs and NNRTIs resistant HIV [21,25,26]. Moreover, the frequent mutations affecting proteinase inhibitors (PIs) antiretroviral drugs are I54V, M46I, and V82A [21,28,29,31,32].

In resource-limited settings, where laboratory testing methods are not easily affordable, epidemiological data are limited. In Africa, there are studies reporting the HIVDR profiles though regional differences are noted, as much of the available evidence are from the South African region of the continent. Few systematic reviews reporting HIVDR for specific populations, and for some of the drug resistance patterns in Africa [34] and Sub-Saharan African [35,36] have been published. Boerma et al reported the magnitude of pretreatment HIVDR was 42.7% among PMTCT-exposed children and 12.7% among PMTCT-unexposed children [36]. Moreover, Ssemwanga et al reported the estimated magnitude of both transmitted and acquired HIVDR from 2001–2011 in Africa and found that the prevalence was 10.6% [34]. Therefore, to better understand the pattern, variation in geographic distribution and type of HIVDR mutations in the African continent, a comprehensive systematic review needs to be conducted. This systematic review and meta-analysis will provide the overall HIVDR magnitude and its associated factors for PLWH who are ART-naïve, on first-line ART, on second-line ART and treatment failure. It will also provide a pooled estimate of HIVDR mutations in Africa.

## Objectives

### General objective

The main objective of this systematic review and meta-analysis will be to estimate the magnitude of HIVDR and HIVDR mutations, and identify factors associated with HIVDR among PLWH in Africa.

### Specific objectives

The specific objectives of this Systematic review protocol are:

- To estimate the magnitude of transmitted HIVDR in people living with HIV/AIDS

- To estimate the magnitude of acquired HIVDR in people living with HIV/AIDS

- To estimate the frequency of HIVDR mutations in people living with HIV/AIDS

- To identify factors associated with HIVDR in PLWH

## Methods

### Design and protocol registration

This systematic review and Meta-analysis protocol is designed in accordance with the updated Preferred reporting Items for Systematic Review and Meta-analysis protocol (PRISMA-P 2015) [37]. It is a systematic review and meta-analysis of quantitative studies. The protocol has

been registered at Prospero, an international database for prospective register of systematic reviews with Registration number: CRD42021266703.

## Eligibility criteria

**Inclusion criteria.** The systematic review will include the following studies

- Studies reported HIVDR that used objective-based criteria to define drug resistance. In this case, studies that employed molecular techniques to identify the HIVDR mutations will be included in the review.

- Studies done using observational design (prospective cohort, retrospective cohort, cross-sectional, retrospective, case-control) and experiential design (clinical trial) will be included in the review.

**Exclusion criteria.** Studies with the following characteristics will be excluded from the systematic review:

- Predicted HIVDR using Mathematical or statistical prediction or forecasting Models

- Commentary, case report, case series and editorial notes,

- Published in languages other than English

## Outcomes

The review has three main outcomes:

1. The pooled prevalence of HIVDR which is defined as the presence of HIVDR mutations in one or more of the genes encoding HIV Reverse transcriptase, protease, and ligase leading to resistance to ART drugs. The magnitude of HIVDR can be calculated by diving the PLWH having one or more HIVDR mutations to the total number of PLWH.

2. The proportion of HIVDR mutations known to have a strong association with NRTIs, NNRTIs and PIs resistance. This can be calculated by dividing the frequency of individual mutation by the total frequency of HIVDR mutations identified.

3. Factors associated with HIVDR that are reported in the form of either odds ratio (OR) hazard ratio (HR).

   The review has also secondary outcomes

1. The proportion of transmitted HIVDR among PLWH who are ART–naive with no history of ARV drug exposure. It can be calculated by dividing the PLWH having one or more HIVDR mutation(s) to the total number of PLWH who are ART-naïve

2. The proportion of acquired HIVDR among PLWH who took or were taking ART medication. It can be calculated by diving the number of PLWH having one or more HIVDR mutation(s) to the total number of PLWH who took or were taking ART medication.

3. The proportion of HIVDR among PLWH who took or were taking first-line ART medication

4. The proportion of HIVDR among PLWH who took or were taking second-line ART medication

## Searching strategy

A comprehensive literature search will be conducted in PubMed/Medline, HINARI, SCOPUS, EMBASE, CINAHL, Web of Sciences, and Cochrane electronic databases. Article published between 2000 and 30 October 2021 will be included. The reason why we began the searching from 2000 is that most of the African countries gave emphasis to HIV prevention and care service as a response to Millennium Development Goals(MDG), endorsed in 2000, particularly MDG 6—combating HIV/AIDS [38]. The search terms will be developed in accordance with the Medical Subject Headings (MeSH) thesaurus using the following a combination of key terms: "HIV" OR "AIDS" OR "human immunodeficiency virus" OR "acquired immunodeficiency syndrome" AND "Pretreatment" OR "Primarily" OR "Initial" OR "Transmitted" OR "Acquired" AND "Drug resistan*" OR "Drug resistance mutation*" OR "Drug resistance genotyp*" AND "Africa" OR "Africa* continent" OR "Africa* region".

African Journal Online (AJOL) repository will be used to search articles. For relevant grey literature, Google Scholar and Google will be used for searching. In addition, conference papers/abstracts will be searched in abstract databases such as the International AIDS Society conferences, conference on Retroviruses and Opportunistic Infections, International HIV Drug Resistance Workshop, and Workshop on HIV Pediatrics. Additional literature will also be searched in HIV/AIDS-related repositories. Reference lists of retrieved articles will be used to identify studies that are not retrieved from databases searched.

## Study selection and quality appraisal

The retrieved articles will be imported to EndNote X7 (Thomson Reuters, New York, USA) bibliographic software. The literature will be screened based on title/abstract by two authors (MMG and HAG) independently after the duplicate articles are removed. Differences will be resolved through discussion. In case of disagreement after discussion, the decision will be made by the third author (PRW). After screening, a full-text appraisal will be done for articles deemed relevant at the screening stage for inclusion in the systematic review and meta-analysis.

The methodological quality of articles will be assessed using Joanna Briggs Institute (JBI) critical appraisal tools depending on the type of study design used by the original studies being appraised: prevalence studies [39], analytical cross-sectional studies [40], Cohort studies [41] and Case-control studies [42]. Similarly, the discrepancies during critical appraisal will be solved through discussion and if not by the decision of the third author/reviewer. Covidence, a systematic reviews production tool, will be used for title/abstract screening, full-text screening, data abstraction and quality assessment.

## Data extraction

For articles eligible to be included in the systematic review, data extraction will be done using an extraction sheet. The following information will be extracted: name of the first author, year of publication, the median year of sample collection/study, country, study design, median/mean age of the study participants, population group (pediatrics/adult), ART status (ART-naïve, on first-line ART, on second-line ART), treatment outcome, genotypic resistance testing methods used, number of PLWH who develop HIVDR, total sample size, frequency of each HIVDR mutations identified, frequency of NRTIs resistance mutations, frequency of NNRTIs resistance mutations, frequency of PIs resistance mutations, and frequency of mutations with dual- and triple- ART class resistance. In addition, the sociodemographic and clinical data will be extracted. The extracted data will be recorded in a customized JBI data extraction sheet.

## Data analysis and interpretation

The data will be exported to and analyzed by Stata version 14 (StataCorp, Texas, USA). The magnitude of heterogeneity between the included studies will be qualitatively assessed by Funnel plot, and quantitatively measured by an index of heterogeneity (I-squared statistics) [43]. I-squared value of 25, 50 and 75% are assumed to represent low, medium, and high heterogeneity, respectively. Because of the expected heterogeneity, the magnitude of HIVDR, the proportion of each HIVDR mutation, and measure of association (OR or HR) will be pooled using a random-effects model. Freeman–Tukey arcsine square root transformation will be used to stabilize the variance of the raw proportions. Sub-group analysis will be conducted considering the following grouping variables: ART status, treatment outcome, sub-continent, population (adult versus pediatrics) and the type of HIVDR. Sensitivity analysis will also be done to the influence of individual studies on the pooled estimate. To see the pattern of HIVDR over time, meta-regression will be done by median year of sampling per study. Publication bias will be assessed busing egger's statistics. In case of publication bias, Trim and Fill analysis will be conducted to overcome a small-study effect.

## Discussion and conclusion

HIV drug resistance is a global concern both in developed and developing nations. In resource-limited setting particularly in African countries where adherence to ART and early initiation of ART is low, it is shown to be the challenge of HIV/AIDS care [44,45]. Based on the survey reports, WHO warns that HIVDR is a growing threat that undermine the global progress in treating and preventing HIV infection unless effective actions are taken [36]. It limits the success of ART and leads to premature treatment failure. It would be a burden for HIV/AIDS care as the drug resistant strains of HIV will further be transmitted to the newly infected patients. To tackle the challenges posed HIVDR, WHO set a global action plan with five strategic objectives, and one of the objective is obtaining data on HIVDR and HIV service quality [46]. Reliable data on prevalence, trend associated factors of HIVDR play a critical role in informing the global, regional, and national policy decisions on ART and HIV service delivery. Moreover, the Joint United Nations Program on HIV/AIDS (UNAIDS) [47] and WHO have set goal of ending the AIDS pandemic as a public health threat by 2030 [47]. Therefore, this systematic review will provide data important to track progress in achieving the set objectives and goals. Besides, the data generated will inform policymakers to revise antiretroviral policies and guidelines. For the clinician, it will guide to choose the ART regimens to achieve optimal treatment outcome.

The UNAIDS transited from its 2020's goal of 90-90-90 to the 2030's goals of '95-95-95' which aimed at achieving 95% of PLWH will know HIV their status, 95% of people diagnosed with HIV infection will receive ART, and 95% of people accessing ART will attain viral suppression [47]. To achieve this ambitious plan, particularly the third 95% target, data on HIVDR and its predictors will have a role to track the progress towards achieving the target.

## Amendment of the protocol

If there is a need for amendment beyond what has described in the protocol during the final reporting of the systematic review and meta-analysis, the authors will describe the reason(s) for the amendments made.

## Strength and limitation

### Strengths

This systematic review will have strengths. Firstly, it will provide a compressive estimate of HIVDR in Africa; secondly, it will also provide evidence on the genotypic profile of

HIVDR mutations; and thirdly, it will identify factors associated with HIVDR among PLWH in Africa.

## Limitations

This systematic review will have some limitations. As the estimation will be done for the African continent, there might be an expected high level of heterogeneity across the study setting as the result of differences in genotyping technique, the population characteristic and disease distribution. In some of the sub-continents, the availability of original studies reporting HIVDR might be limited. This will unduly influence the generalizability of the estimates.

## Supporting information

**S1 File. PRISMA-P 2015 checklist.**
(DOCX)

**S2 File. Search strategy for major databases.**
(DOC)

## Acknowledgments

The authors would like to thank the Flinders University for authorizing full access of the library and the Flinders University librarians for their valuable assistance in constructing search strategies.

## Author Contributions

**Conceptualization:** Mulugeta Melku, Hailay Abrha Gesesew, Paul R. Ward.

**Data curation:** Mulugeta Melku, Paul R. Ward.

**Investigation:** Mulugeta Melku, Hailay Abrha Gesesew, Paul R. Ward.

**Methodology:** Mulugeta Melku, Hailay Abrha Gesesew, Paul R. Ward.

**Project administration:** Mulugeta Melku, Hailay Abrha Gesesew, Paul R. Ward.

**Resources:** Mulugeta Melku, Hailay Abrha Gesesew, Paul R. Ward.

**Software:** Mulugeta Melku, Hailay Abrha Gesesew, Paul R. Ward.

**Supervision:** Hailay Abrha Gesesew, Paul R. Ward.

**Validation:** Mulugeta Melku, Hailay Abrha Gesesew, Paul R. Ward.

**Visualization:** Mulugeta Melku, Hailay Abrha Gesesew, Paul R. Ward.

**Writing – original draft:** Mulugeta Melku.

**Writing – review & editing:** Hailay Abrha Gesesew, Paul R. Ward.

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
