## [Decision Letter · Decision Letter 0]

17 Mar 2022

PONE-D-21-32189Magnitude and predictors of HIV-Drug Resistance in Africa: a protocol for Systematic Review and Meta-analysisPLOS ONE

Dear Dr. Melku,

Thank you for submitting your manuscript to PLOS ONE. After careful consideration, we feel that it has merit but does not fully meet PLOS ONE’s publication criteria as it currently stands. Therefore, we invite you to submit a revised version of the manuscript that addresses the points raised during the review process.

We look forward to receiving your revised manuscript.

Kind regards,

Felix Bongomin, MB ChB, MSc, MMed (Int Med), FECMM

Academic Editor

PLOS ONE

Journal Requirements:

Additional Editor Comments:

Please address the several minor issues raised by the reviewers

Reviewers' comments:

Reviewer's Responses to Questions

**Comments to the Author**

1. Does the manuscript provide a valid rationale for the proposed study, with clearly identified and justified research questions?

Reviewer #1: Yes

Reviewer #2: Yes

2. Is the protocol technically sound and planned in a manner that will lead to a meaningful outcome and allow testing the stated hypotheses?

Reviewer #1: Yes

Reviewer #2: Yes

3. Is the methodology feasible and described in sufficient detail to allow the work to be replicable?

Reviewer #1: Yes

Reviewer #2: Yes

4. Have the authors described where all data underlying the findings will be made available when the study is complete?

Reviewer #1: Yes

Reviewer #2: Yes

5. Is the manuscript presented in an intelligible fashion and written in standard English?

Reviewer #1: No

Reviewer #2: Yes

6. Review Comments to the Author

You may also provide optional suggestions and comments to authors that they might find helpful in planning their study.

Reviewer #1: Thank you for the efforts towards research and academia. However, I noted the following;

1. Introduction, 4th paragraph, 10th line, 1st word: "sab-saharan" instead of sub-saharan

2. Data extraction, 3rd last line, 5th word: "mutilations" instead of mutations

3. Discussion, 1st paragraph , 6th last line Joint United Nations Program on HIV/AIDS -"USAIDS" instead of UNAIDS

1st paragraph last line ART "regiments" instead of ART regimens

4. Limitations, 2nd last line "HIDR" instead of HIVDR

Reviewer #2: 1. Multiple claims in the manuscript need citations. For example (a) The resistance to first-line ART drugs would also affect effectiveness of second-line ART drug, (b) In Africa, its prevalence has been raising. According to the 2019 WHO report, in six African countries that participated in national HIVDR surveillance, the prevalence is high in children and adults, (c) The world health organization (WHO) reported that up to 26% of people initiating treatment are infected with a virus carrying resistance to first-line antiretroviral drugs, (d) At the end of 2020, 37.6 million (95% uncertainty level: 30.2 - 45 million) people were living with HIV, of which more than two-thirds were in the African WHO region.

2. Some minor typos. Change sab-sahara to sub-sahara, also change 42,7% to 42.7%. Revise the whole manuscript to remove some typos errors

7. PLOS authors have the option to publish the peer review history of their article (what does this mean?). If published, this will include your full peer review and any attached files.

Reviewer #1: **Yes: **Kaggwa Henry

Reviewer #2: No

---

## [Author Response · Author response to Decision Letter 0]

25 Mar 2022

Authors Response to Reviewers’ Comments (PONE-D-21-32189) 

PONE-D-21-32189

Magnitude and predictors of HIV-Drug Resistance in Africa: a protocol for Systematic Review and Meta-analysis

PLOS ONE

Date: 19/03/2022

Response to Editor

Additional Editor Comments:

Please address the several minor issues raised by the reviewers

Reviewers' comments:

Reviewer's Responses to Questions

Response: Thank you for the timely editorial assessment. We addressed the review comments raised by reviewers. 

Comments to the Author

1. Does the manuscript provide a valid rationale for the proposed study, with clearly identified and justified research questions?

Reviewer #1: Yes

Reviewer #2: Yes

Response: Thank you 

 2. Is the protocol technically sound and planned in a manner that will lead to a meaningful outcome and allow testing the stated hypotheses?

Reviewer #1: Yes

Reviewer #2: Yes

Response: Thank you 

3. Is the methodology feasible and described in sufficient detail to allow the work to be replicable?

Reviewer #1: Yes

Reviewer #2: Yes

Response: Thank you 

4. Have the authors described where all data underlying the findings will be made available when the study is complete?

Reviewer #1: Yes

Reviewer #2: Yes

Response: Thank you 

5. Is the manuscript presented in an intelligible fashion and written in standard English?

Reviewer #1: No

Response: The protocol is edited for clarity and language 

Reviewer #2: Yes

Response: Thank you 

Response to reviewer #1

Response: The authors would like to thank the reviewers for the scientific and technical comments which would improve the protocol.

Reviewer #1: Thank you for the efforts towards research and academia. However, I noted the following; 

1. Introduction, 4th paragraph, 10th line, 1st word: "sab-saharan" instead of sub-saharan

Response: Thank you for the prudence in reviewing the protocol. We corrected as suggested 

2. Data extraction, 3rd last line, 5th word: "mutilations" instead of mutations

Response: Thank you for the critical review. We corrected as suggested 

3. Discussion, 1st paragraph, 6th last line Joint United Nations Program on HIV/AIDS -"USAIDS" instead of UNAIDS

1st paragraph last line ART "regiments" instead of ART regimens

Response: Thank you for the valuable checking of typos. We corrected as suggested 

4. Limitations, 2nd last line "HIDR" instead of HIVDR

Response: Thank you for the valuable checking of typos. We corrected as suggested

Response to reviewer #2

Response: The authors would like to thank the reviewers for the scientific and technical comments which would improve the protocol.

Reviewer #2: 

1. Multiple claims in the manuscript need citations. For example 

(a) The resistance to first-line ART drugs would also affect effectiveness of second-line ART drug.

Response: Thank you.

We supported this argument with a subsequent description thereafter citing references stepwise (Ref 19 -24). We did so because we believed that once the argument is supported by subsequent statement(s), the use of citation for the umbrella sentence is not a must. But, for this revision of the manuscript, we cited references as you suggested to do so (see Paragraph #3). 

(b) In Africa, its prevalence has been raising. According to the 2019 WHO report, in six African countries that participated in national HIVDR surveillance, the prevalence is high in children and adults.

Response: Thank you. 

Two additional references supporting the argument are added in this version of the protocol. Besides, those citation aftermath of these sentence support the arguments that the authors stated (see paragraph #3 of the introduction section).

(c) The world health organization (WHO) reported that up to 26% of people initiating treatment are infected with a virus carrying resistance to first-line antiretroviral drugs, 

Response: The reference #10 cited after the immediate sentence is for both statements. “The …….WHO reported that up to 26%..........carrying resistance to first-line antiretroviral drugs. The level of HIVDR is higher ………….. up to 69% (10).” If it is a must cite the reference for both sentence separately, we can and it is so in this version. In addition to reference #10, we supported the argument with reference #11 cited in this revision (see paragraph #3 of introduction). . 

(d) At the end of 2020, 37.6 million (95% uncertainty level: 30.2 - 45 million) people were living with HIV, of which more than two-thirds were in the African WHO region.

Response: Thank you. The evidence is obtained from ‘WHO HIV/AIDS key facts’ website cited in the document after taking the preceding statement next to this sentence. “……in African WHO Region. Moreover, one and half million people were newly infected, and more than half a million people died from HIV-related causes in 2020 (1)”. If it is a must to cite evidence for every sentence, we can cite reference #1 for this sentence, and we did so. 

2. Some minor typos. Change sab-sahara to sub-sahara, also change 42,7% to 42.7%. Revise the whole manuscript to remove some typos errors. 

Response:

Thank you. We have made corrections and editions based on the reviewers’ comments. 

Additional information to editor & reviewers:

 In this revised version of the protocol, we included supplementary file which showed the search strategies being used for article retrieval from major databases (see S3 file). 

Sincerely,

Mulugeta Melku, Corresponding authors

---

## [Editor Report · Decision Letter 1]

4 Apr 2022

Magnitude and predictors of HIV-Drug Resistance in Africa: a protocol for Systematic Review and Meta-analysis

PONE-D-21-32189R1

Dear Dr. Melku,

We’re pleased to inform you that your manuscript has been judged scientifically suitable for publication and will be formally accepted for publication once it meets all outstanding technical requirements.

Kind regards,

Felix Bongomin, MB ChB, MSc, MMed, FECMM

Academic Editor

PLOS ONE
---

## [Editor Report · Acceptance letter]

7 Apr 2022

PONE-D-21-32189R1 

Magnitude and predictors of HIV-Drug Resistance in Africa: a protocol for Systematic Review and Meta-analysis 

Dear Dr. Melku:

I'm pleased to inform you that your manuscript has been deemed suitable for publication in PLOS ONE. Congratulations! Your manuscript is now with our production department. 

Kind regards, 

on behalf of

Dr. Felix Bongomin 

Academic Editor

PLOS ONE